# Differential Responses of Human iPSC-Derived Microglia to Stimulation with Diverse Inflammogens

**DOI:** 10.3390/cells14211687

**Published:** 2025-10-28

**Authors:** Chiara Wolfbeisz, Julian Suess, Nadine Dreser, Heidrun Leisner, Markus Brüll, Madeleine Fandrich, Nicole Schneiderhan-Marra, Oliver Poetz, Thomas Hartung, Marcel Leist

**Affiliations:** 1In Vitro Toxicology and Biomedicine, Chair inaugurated by the Doerenkamp-Zbinden Foundation, University of Konstanz, 78464 Konstanz, Germany; 2Natural and Medical Sciences Institute, University of Tuebingen, 72074 Tuebingen, Germany; 3SIGNATOPE GmbH, 72770 Reutlingen, Germany; 4Center for Alternatives to Animal Testing (CAAT)-Europe, University of Konstanz, 78464 Konstanz, Germany; 5Doerenkamp-Zbinden Chair for Evidence-Based Toxicology, Johns Hopkins University, Baltimore, MD 21205, USA; 6Center for Alternatives to Animal Testing (CAAT), Johns Hopkins University, Baltimore, MD 21205, USA

**Keywords:** (neuro-)inflammation, TLR, TNFα, astrocytes, cytokine release, transcriptome changes

## Abstract

Human microglia are central regulators and actors in brain infections and neuro-inflammatory pathologies. However, access to such cells is limited, and studies systematically mapping the spectrum of their inflammatory states are scarce. Here, we generated microglia-like cells (MGLCs) from human induced pluripotent stem cells and characterized them as a robust, accessible model system for studying inflammatory activation. We validated lineage identity through transcriptome profiling, revealing selective upregulation of microglial signature genes and enrichment of microglia/macrophage-related gene sets. MGLCs displayed distinct morphologies and produced stimulus- and time-dependent cytokine secretion profiles upon exposure to diverse inflammatory stimuli, including pro-inflammatory cytokines (TNFα, interferon-γ) and agonists of the Toll-like receptors TLR2 (FSL-1), TLR3 (Poly(I:C)), TLR4 (lipopolysaccharide, LPS), and TLR7 (imiquimod). Transcriptome profiling and bioinformatics analysis revealed distinct activation signatures. Functional assays demonstrated stimulus-specific engagement of NFκB and JAK-STAT signaling pathways. The shared NFκB nuclear translocation response of TLR ligands and TNFα was reflected in overlapping transcriptome profiles: they shared modules (e.g., oxidative stress response and TNFα-related signaling) identified by weighted gene co-expression network analysis. Finally, the potential consequences of microglia activation for neighboring cells were studied on the example of microglia-astrocyte crosstalk. The capacity of MGLC supernatants to stimulate astrocytes was measured by quantifying astrocytic NFκB translocation. MGLCs stimulated with FSL-1, LPS, or Poly(I:C) indirectly activated astrocytes via a strictly TNFα-dependent mechanism, highlighting the role of soluble mediators in the signal propagation. Altogether, this platform enables a dissection of microglia activation states and multi-parametric characterization of subsequent neuroinflammation.

## 1. Introduction

Microglia, the local macrophage population of the brain, react to a large variety of inflammatory stimuli. The different responses of human microglia require thorough characterization to better understand their respective contributions to toxicological insults and neurodegenerative states [1,2,3,4,5,6]. Microglia make up 0.5–17% of the human brain cells, depending on the tissue region, developmental stage, and several other factors [7,8]. Their multiple branched processes or ramifications constantly survey and modify the central nervous system (CNS) environment, in particular the synapses [9,10,11].

In the past, microglia have been considered to exist in at least three states: inactive, pro-inflammatory, and anti-inflammatory [12]. However, they are never truly quiescent, even under healthy conditions. They maintain brain homeostasis and play important functions during developmental processes [10]. Under pro-inflammatory conditions, they secrete inflammatory cytokines and chemokines and phagocytose unwanted material, e.g., pathogens, debris, and synapses. In contrast, microglia can also dampen acute inflammation and exhibit highly diverse functions and morphologies [13]. The concept of multiple activation states is in line with similar observations in other macrophage populations [14,15,16]. This concept is paralleled by descriptions of diverse activation states in other glial cell types, such as astrocytes [17,18,19].

The diversity of microglial response patterns is created on many levels. Microglia express a broad repertoire of PRRs (pattern recognition receptors). For instance, TLRs (Toll-like receptors) are activated by a large panel of PAMPs (pathogen-associated molecular patterns), which are found in/on bacteria, viruses, and other potential pathogens [20,21,22]. Further triggers or modulators of microglia activation are blood-derived cytokines/chemokines or signals from neighboring CNS cells [23,24]. TLR4 is one of the best-studied PRRs. It is activated by lipopolysaccharide (LPS), a major component of Gram-negative bacterial cell walls [25,26,27]. LPS has been known as an inflammogen for over a century, and it has been used as archetypical trigger of inflammation in numerous in vivo and in vitro models. It is also the most frequently used trigger of microglia [28].

Other TLR ligands that stimulate microglia include FSL-1 (FSL), a diacylated lipopeptide binding to TLR2/6; imiquimod (IQM), a synthetic immune response modifier binding to TLR7; and Poly(I:C) (I:C), a synthetic analog of viral double-stranded RNA, triggering antiviral responses through Toll-like receptor 3 [29]. Also, the cytokines tumor necrosis factor-α (TNF) and interferon-γ (IFN) are well-known microglia modulators. TLR signaling leads to inflammatory responses mainly through activation of the nuclear factor kappa-light-chain-enhancer of activated B cells (NFκB), which acts as a transcription factor and by triggering the mitogen-activated protein kinase (MAPK) pathway [14,30]. TNF receptors also trigger inflammation via NFκB activation. IFN uses a different signal transduction cascade. It links to JAK kinases, which activate STAT transcription factors, and thereby the transcription of a panel of target genes [31,32,33,34,35]. In all cases, microglia activation results in the release of cytokines (e.g., TNF, IL-1β, and IL-6), chemokines, and other mediators. However, the distinct release patterns may differ largely between stimuli [1,2,3,4,5,6].

Microglia generally act as first line of defense in the CNS, by recognizing pathogens and passing this information onto other cells [36,37,38,39]. Their activation and signaling not only enhance their own anti-pathogenic functions but also influence neighboring cell types, such as neurons and astrocytes. Astrocytes then take over important tasks, such as the recruitment of blood cells, stabilization of the tissue, and antigen presentation [40]. Upon activation, microglia can secrete interleukin-1 alpha (IL-1α), TNF, and complement component C1q, molecules that have been shown to drive astrocytes towards a reactive and pro-inflammatory phenotype [41]. In response, astrocytes signal via their internal inflammatory cascades, e.g., NFκB. This leads to the production of additional mediators [17,39,41].

A binary description of microglia and astrocyte activation states as M1/M2 or A1/A2 may be an oversimplification [15,18,42,43]. The diverse glial activation states reflect the complex interplay of signals from the environment, leading to a range of functional profiles with consequences for cell fate and behavior, e.g., transcriptional changes, morphology, and cytokine release. The complexity of these activation states and their precise roles in various physiological and pathological conditions is most likely insufficiently modeled by relying on a single (LPS) or very few inflammatory stimuli.

Moreover, a large amount of knowledge on microglia has been generated in rodent models. However, there are significant species differences between rodent and human immune systems, which also extend to microglial functions [44,45,46]. For instance, large response differences have been observed for cytokine release, NO production, translocator protein (TSPO), metabolic reprogramming, and phagocytosis [47,48,49,50]. Such differences highlight the limitations of directly translating findings from rodent models to human biology and the importance of providing and studying human models [2,5,51]. A major challenge in studying human microglia is the scarcity of primary human brain cells. To tackle this issue, the field has increasingly turned to the generation of the required cell types from induced pluripotent stem cells (iPSCs). Numerous protocols for generating iPSC-derived microglia have been published [51,52,53,54]. They provide a homogeneous and abundant source of cells for research and recapitulate key features of primary human microglia [51,52,53].

We aimed here at establishing a MGLC-based experimental platform that allowed a phenotypic and functional characterization of activation state diversity. This systematic approach used cell activation by the cytokines TNF and IFN as well as by the TLR ligands LPS, I:C, FSL, and IQM. Gene expression signatures, signaling pathways, and cytokine secretion were explored. We were especially interested in resolving the temporal evolution of the responses in more detail. The crosstalk with astrocytes was explored as an example of the functional consequence of MGLC activation.

## 2. Materials and Methods

### 2.1. Stem Cell Culture

The induced pluripotent stem cells (iPSCs) (Si28, EPITHELIAL-1, #IPSC0028; Sigma-Aldrich, Traufkirchen, Germany) were cultured in mTesR1 medium (StemCell Technologies, Vancouver, BC, Canada) on Matrigel-coated (Corning, Somerville, MA, USA) tissue culture dishes (60 mm, Sarstedt, Nümbrecht, Germany) and passaged at 80–90% confluency using EDTA. Cells were incubated at 37 °C and 5% CO_2_, and medium was changed daily.

### 2.2. Factory Differentiation and PreMacs Generation

The differentiation protocol for microglia-like cells (MGLCs) from human induced pluripotent stem cells (iPSCs) was adapted from [51,55,56]. In brief, Aggrewell 800 plates (StemCell Technologies, Vancouver, BC, Canada) were prepared by addition of 0.5 mL of anti-adherence rinsing solution (AARS; StemCell Technologies, Vancouver, BC, Canada) per well and centrifuged at 2000× *g* for 5 min in a swinging bucket rotor to remove bubbles from the microwells. The AARS was removed, 1.5 mL mTeSR1 was added per well, and the plate was centrifuged again. After removal, 1 mL of mesoderm induction medium (mTeSR1 medium supplemented with 50 ng/mL human bone morphogenetic protein 4 (hBMP4; Peprotech, Hamburg, Germany), 50 ng/mL human vascular endothelial growth factor (hVEGF; Miltenyi Biotech, Bergisch Gladbach, Germany), and 20 ng/mL human stem cell factor (hSCF; Miltenyi Biotech, Bergisch Gladbach, Germany) containing 10 µM rho-associated protein kinase inhibitor (ROCKi; Y-27632; Tocris, Bristol, UK)) was added per well, and plates were centrifuged again. ROCKi was only added for the first day. The iPSCs were washed with 1 mL Advanced DMEM/F-12 (Gibco by Thermo Fisher Scientific, Waltham, MA, USA) and then incubated with 1 mL Accutase (PanBiotech, Aidenbach, Germany) for 3–5 min at 37 °C. Accutase was stopped with 9 mL Advanced DMEM/F-12, and cells were harvested by centrifugation at 500× *g* for 3 min. The cell number of the iPSC suspension was adjusted to 4 × 10^6^ cells per ml (in mesoderm induction medium containing 10 µM ROCKi). Next, 1 mL of the cell suspension was added per each well in an Aggrewell plate into the existing medium (2 mL final volume). The Aggrewell plate was centrifuged at 100× *g* for 3 min and checked by microscopy for an even distribution of the cells. Aggrewell plates were incubated for seven days at 37 °C and 5% CO_2_ with daily feedings by replacing 1 mL of the 2 mL in each well twice with mesoderm induction medium to induce embryoid bodies (EBs). On day 7, EBs were dislodged by gently pipetting up and down and transferred to a 40 µm cell strainer (EBs were caught by the filter). The filter was inverted, and EBs were washed off using 5 mL of “factory medium” consisting of X-VIVO 15 medium (Lonza, Basel, Switzerland) supplemented with 2 mM Glutamax, 100 U/mL penicillin–streptomycin, and 50 µM mercaptoethanol (all three from Gibco by Thermo Fisher Scientific, Waltham, MA, USA), M-CSF (100 ng/mL), and IL-3 (25 ng/mL) (both from Miltenyi Biotech, Bergisch Gladbach, Germany). EBs from one Aggrewell were seeded into two T75 flasks containing 10 mL factory medium. T75 flasks were coated with 7.5 mL growth factor reduced Matrigel (GFR-Matrigel; Gibco by Thermo Fisher Scientific, Waltham, MA, USA) diluted 1:30 in Advanced DMEM/F12.

After seven days of EB plating, 5.5 mL factory medium was added. In the following weeks, factories received weekly fresh factory medium as follows: after 14 days of factory initiation, 7.5 mL was added; after 18 days, half medium exchange (10 mL); then a full medium exchange twice per week (20 mL).

### 2.3. Harvest of Macrophage Precursor Cells (PreMacs)

Macrophage precursor cells (PreMacs) were released into the factory supernatant. The removed medium was centrifuged at 400× *g* for 4 min to harvest a PreMacs pellet. After the second full medium exchange, flow cytometry analysis of PreMacs was performed to ensure the quality of differentiation. PreMacs were either put back into factories once per week or used immediately for further differentiation in MGLCs. Factories can be maintained for up to day 100–120 of differentiation. When PreMacs production or marker expression dropped, factories were discarded.

### 2.4. Maturation to Microglia-like Cells (MGLCs)

Harvested PreMacs were re-suspended in “MGLC differentiation medium” (Advanced DMEM/F12 with 1 × N2 supplement (Gibco by Thermo Fisher Scientific, Waltham, MA, USA), 100 U/mL penicillin–streptomycin, 2 mM Glutamax, 50 mM β-mercaptoethanol, 100 ng/mL rhIL34, and 10 ng/mL rhGM-CSF (both from Miltenyi Biotech, Bergisch Gladbach, Germany)) and plated at densities between 100,000–133,000 cells/cm^2^ on 1 µg/mL fibronectin (Sigma-Aldrich, Traufkirchen, Germany) in DPBS-coated tissue culture dishes. Fibronectin coating was performed at 37 °C and 5% CO_2_ overnight, and coating was removed directly before plating the cells. After 4 days of culture (DoC), MGLCs were replated and cultured for at least additional three days (until DoC7) at 37 °C and 5% CO_2_ before the experiment (unless stated differently). To detach the MGLCs, cells were incubated with 1 mL Accutase for 5 min at 37 °C. Accutase was stopped with 3–4 × volume of Advanced DMEM/F12, and cells were harvested by centrifugation at 500× *g* for 4 min. MGLCs were cultured under the same conditions as the freshly harvested PreMacs. If cells were not replated, a half medium exchange was performed every three to four days.

### 2.5. Differentiation and Culture of iPSC-Derived Astrocytes

Astrocytes were differentiated in-house as previously described [17] and were used for experiments between 64 and 130 days of differentiation. Briefly, astrocytes were frozen in astrocyte differentiation medium (DMEM/F12 with 1 × N2 supplement, 1 × B27 supplement (both Gibco by Thermo Fisher Scientific, Waltham, MA, USA), 2 mM L-glutamine (Sigma-Aldrich, Traufkirchen, Germany), and supplemented with 1% fetal bovine serum (FBS) (PAA Laboratories, Cölbe, Germany)) with 10% dimethylsulfoxide (DMSO, Merck, Darmstadt, Germany). After thawing, cells were immediately transferred into 10 mL DMEM/F12 and centrifuged at 500× *g* for 4 min. Astrocytes were re-suspended in astrocyte differentiation medium and seeded at a density of 60,000 cells/cm^2^ on Matrigel-coated tissue culture dishes and split once per week. Matrigel coating (1:40 in DMEM/F-12) was incubated for 30 min at 37 °C and 5% CO_2_, and unadsorbed Matrigel was removed directly before seeding the cells. For reseeding, cells were detached by incubation with Accutase for 5–8 min at 37 °C, 5% CO_2_. Cells were completely washed off with DMEM/F-12, and the cell suspension was centrifuged at 500× *g* for 4 min. The supernatant was removed, and the cell pellet was re-suspended in astrocyte differentiation medium. Cells were seeded as described above.

For the quantification of NFκB translocation, 96-well plates were coated with 43 µg/mL poly-L-ornithine hydrobromide (PLO; Sigma-Aldrich, Traufkirchen, Germany) in Milli-Q H_2_O. After incubation at 37 °C overnight, plates were washed three times with Milli-Q H_2_O. Then plates were coated with 1 µg/mL laminin and 1 µg/mL fibronectin (both from Sigma-Aldrich, Traufkirchen, Germany) in Milli-Q H_2_O at 37 °C overnight. The coating was removed immediately before seeding the cells at a density of 40,000 cells/cm^2^.

### 2.6. Inflammogens and Treatments

The following inflammogens and treatments were used throughout the whole study, if not otherwise stated: 100 ng/mL lipopolysaccharide (LPS from *Escherichia coli* O55:B5; Sigma-Aldrich, Traufkirchen, Germany), 10 µg/mL polyinosinic:polycytidylic acid (Poly(I:C) = I:C; high-molecular weight; Invivogen, San Diego, CA, USA), 100 ng/mL FSL-1 (FSL from *Mycoplasma salivarium*; Pam2CGDPKHPKSF; Invivogen, San Diego, CA, USA), 10 µg/mL Imiquimod (R837) (IQM; Invivogen, San Diego, CA, USA), 10 ng/mL TNFα (TNF; R&D Systems, Mineapolis, MN, USA), 20 ng/mL IFNγ (IFN; Peprotech, Hamburg, Germany), 100 µM dexamethasone (DEX; Sigma-Aldrich, Traufkirchen, Germany). The concentrations of the inflammogens and treatments used in this study were chosen based on a previous study [57] and are consistent with those in similar studies.

### 2.7. Flow Cytometry

For quality control of the differentiations, flow cytometrical measurements of marker expression of PreMacs were adapted from Gutbier et al. [58]. Briefly, after the first full medium exchange of the factories, PreMacs were analyzed by flow cytometry. Harvested PreMacs were centrifuged at 500× *g* for 4 min and re-suspended in autoMACS running buffer (Miltenyi Biotech, Bergisch Gladbach, Germany). A total of 500,000 PreMacs from the same factory were split into two tubes (one for staining and one as an unstained control). The cells were centrifuged at 500× *g* for 4 min. The supernatant was removed, and the cells were re-suspended in 50 μL Viobility staining solution (diluted 1:100 in DPBS; Miltenyi Biotech, Bergisch Gladbach, Germany) and incubated for 10 min in the dark. After incubation, 1 μL of a pre-mix containing equal parts of the antibodies CD11b, CD14, and CD16 (Miltenyi Biotech, Bergisch Gladbach, Germany) was added to the tube for staining, followed by another 10 min incubation in the dark. AutoMACS running buffer (500 μL) was added before centrifuging the cells at 500× *g* for 4 min. A list of used antibodies can be found in the Appendix A. The supernatant was discarded, and cells were re-suspended in 500 μL cold, freshly prepared buffer 1, following the instructions of the FoxP3 staining buffer set (Miltenyi Biotech, Bergisch Gladbach, Germany). After incubation (30 min, 4 °C, dark), 550 μL of autoMACS running buffer (Miltenyi Biotech, Bergisch Gladbach, Germany) was added, and the tubes were centrifuged (500× *g*, 4 min, room temperature). The supernatant was discarded, and the cells were re-suspended in 100 μL cold buffer 2, also following the instructions of the FoxP3 staining buffer set. For permeabilization, the cells were incubated for 30 min and 4 °C in the dark before adding 0.75 μL of the CD68 antibody to the tube with the stained PreMacs, followed by another incubation (30 min, 4 °C, dark). Subsequently, 500 μL of buffer 2 of the FoxP3 staining buffer set was added, the tubes were centrifuged (500× *g*, 4 min, room temperature), and the cells were re-suspended in 400 μL cold autoMACS running buffer. Flow cytometry was performed with a CytoFLEX LX flow cytometer (Beckman Coulter-9, Brea, CA, USA).

The applied gating strategy was as follows (exemplarily shown in Appendix A): Cells were separated from clumps and debris in the forward–sideward scatter area blot, and doublet exclusion was performed by plotting the width of the forward scatter against its height. Cells negative for the cell death staining (indicating a compromised cell membrane) were selected as viable. These cells were then gated separately for CD11b, CD14, CD16, and CD68. For each sample 30,000 events (cells) were measured.

### 2.8. Scanning Electron Microscopy

For scanning electron microscopy (SEM), 12 mm glass coverslips were coated with 1 µg/mL fibronectin in DPBS in 24-well plates at 37 °C overnight. DoC7 MGLCs were plated at densities of 50,000 cells/cm^2^ and left for 1 h to attach. Samples were prepared as previously described [59]. Briefly, for pre-fixation, half the medium was replaced by 3% formaldehyde and 2% glutaraldehyde in 0.1 M 4-(2-hydroxyethyl)-1-piperazineethanesulfonic acid (HEPES) buffer with 0.09 M sucrose, 0.1 M MgCl_2_, and 0.1 M CaCl_2_ at pH 7.0 for 15 min at room temperature. Subsequent chemical fixation was performed at 4 °C; the fixation of MGLCs was undertaken in the above-noted pre-chilled fixative for 30 min. Samples were washed three times in pre-chilled 0.1 M HEPES buffer with 0.09 M sucrose, 0.1 M MgCl_2_, and 0.1 M CaCl_2_ at pH 7.0 for 5 min each. Then, samples were passed through graded concentrations of pre-chilled 30% and 50% ethanol for 5 min each and of pre-chilled 70% ethanol for 7 min. Dehydration was continued at room temperature in 10% steps of 10 min each and finished in ethanol over a molecular sieve for 10 min twice. Samples were critical-point-dried in CO_2_ in a Leica EM CPD300 (Leica Microsystems, Wetzlar, Germany) and sputter-coated with a 6 nm thick layer of platinum in a Quorum Q150R ES (Quorum technologies, Laughton, UK). Micrographs were produced using a Zeiss Auriga FESEM (Zeiss, Oberkochen, Germany).

### 2.9. Bright Field Imaging

MGLCs on DoC4 were seeded on 1 µg/mL fibronectin in DPBS-coated 96-well plates at a density of 50,000 cells/cm^2^. After three days, MGLCs were stimulated for 48 h with different inflammogens (see above), and images were taken with an Incucyte SX5 (Satorius, Göttingen, Germany).

### 2.10. Treatment of MGLCs for Transcriptome and Cytokine Sample Generation

MGLCs on DoC4 were seeded on 1 µg/mL fibronectin in DPBS-coated 48-well plates at a density of 133,000 cells/cm^2^. On DoC7, a full medium exchange was performed, and MGLCs were treated with medium only or the stimuli listed under inflammogens. The treatment was performed in a reverse manner, and cells were incubated for 18 h, 8 h, and 4 h with the respective treatment. Three (cytokine samples) to four (transcriptome samples) independent biological replicates with one technical replicate each were prepared.

### 2.11. Cytokine Analyses by Multiplexed Microsphere-Based Sandwich Immunoassays

MGLCs were treated as described above. At the respective incubation time, 220 µL of a total volume of 260 µL per well was transferred into Eppendorf tubes and centrifuged at 500× *g* at 4 °C for 5 min. 90 µL was transferred into a separate tube and frozen in liquid nitrogen. Samples were stored at −80 °C and shipped on dry ice for sample analysis.

Cytokine concentrations in supernatants of cell cultures were analyzed as previously published in Ghosh et al. [60] using multiplexed microsphere-based sandwich immunoassays [61]. In short, cytokine concentrations were determined using two multiplex assay panels. Assay panel 1 employed the following capture antibodies: IL-4 (BD Pharmingen, Franklin Lakes, NJ, USA), IL-6 (RnD Systems, Minneapolis, MN, USA), IL-8 (BD Pharmingen, Franklin Lakes, NJ, USA), IL-10 (BD Pharmingen, Franklin Lakes, NJ, USA), IFN (RnD Systems, Minneapolis, MN, USA), CCL-2 (=MCP-1; BD Pharmingen, Franklin Lakes, NJ, USA), MIP-1β (RnD Systems, Minneapolis, MN, USA), and TNF (RnD Systems, Minneapolis, MN, USA), while panel 2 included capture antibodies for IL-1β (RnD Systems, Minneapolis, MN, USA) and IL-1Ra (Invitrogen, Waltham, MA, USA). The capture antibodies were covalently bound to color-coded magnetic polystyrene microspheres (Diasorin Luminex, Austin, TX, USA) as described elsewhere [61]. The supernatants were generally diluted at a ratio of 1:120 for assay panel 1 and 1:20 for panel 2 to match the dynamic assay ranges. 25 µL of calibrator proteins, three biological quality controls and diluted supernatants were incubated together with 25 µL of microsphere mixture, comprising 2000 antibody-coated magnetic microspheres per assay, in a 96-well low binding half-well plate (Corning, NY, USA) at room temperature and 750 rpm for 2h on a plate shaker. The following recombinant proteins were used as calibrators for panel 1: IL-4 (BD Pharmingen, Franklin Lakes, NJ, USA), IL-6 (RnD Systems, Minneapolis, MN, USA), IL-8 (BD Pharmingen, Franklin Lakes, NJ, USA), IL-10 (BD Pharmingen, Franklin Lakes, NJ, USA), IFN (BD Pharmingen, Franklin Lakes, NJ, USA), CCL-2 (=MCP-1; BD Pharmingen, Franklin Lakes, NJ, USA), MIP-1β (RnD Systems, Minneapolis, MN, USA), and TNF (RnD Systems, Minneapolis, MN, USA). Panel 2 used the following recombinant proteins as calibrators: IL-1β (RnD Systems, Minneapolis, MN, USA) and IL-1Ra (RnD Systems, Minneapolis, MN, USA).

After the capture step, excess sample was removed by washing the microspheres three times with 100 µL PBS with 0.05% TWEEN 20 on a magnetic plate (Diasorin Luminex, Austin, TX, USA). The sandwich complexes were formed by incubating with 30 µL detection antibody mix, consisting of the respective biotinylated detection antibodies at room temperature and 750 rpm for 60 min on a plate shaker. The following biotinylated detection antibodies were used for panel 1: IL-4 (BD Pharmingen, Franklin Lakes, NJ, USA), IL-6 (RnD Systems, Minneapolis, MN, USA), IL-8 (BD Pharmingen, Franklin Lakes, NJ, USA), IL-10 (BD Pharmingen, Franklin Lakes, NJ, USA), IFN (RnD Systems, Minneapolis, MN, USA), CCL-2 (MCP-1; BD Pharmingen, Franklin Lakes, NJ, USA), MIP-1β (RnD Systems, Minneapolis, MN, USA), and TNF (RnD Systems, Minneapolis, MN, USA). Panel 2 included capture antibodies for IL-1β (RnD Systems, Minneapolis, MN, USA) and IL-1Ra (Invitrogen, Waltham, MA, USA). Excess detection antibodies were removed by washing the microspheres three times with 100 µL PBS with 0.05% TWEEN 20 on a magnetic bead plate. Finally, the formed sandwich complexes were incubated with 30 µL streptavin–phycoerythrin conjugate (MOSS, Pasadena, MD, USA) at room temperature and 750 rpm for 30 min on a plate shaker. The microtiter plate was transferred to a microsphere reader (Intelliflex Instrument, Luminex, Austin, TX, USA), and fluorescence data were acquired according to the manufacturer’s instructions. Cytokine levels were calculated relating to a 4-paramatric-logarithmic fit using the calibrator results.

### 2.12. Transcriptome Data Generation and Analysis

Transcriptome samples were generated as previously described [17,62] and according to the BioClavis TempO-Seq manual v4. In brief, following the respective differentiation or treatment period, the medium was aspirated, and cells were lysed in enhanced 1× BioSpyder lysis buffer (BioClavis Tech., Glasgow, UK) at a target concentration of 1.5–1.7 × 10^6^ cells/mL (for example, 60 µL/well for inflammatory-stimulated samples). Lysis was carried out at 37 °C for 10 min and included one additional freeze–thaw cycle as recommended by BioClavis. Lysates were stored at −80 °C and shipped on dry ice. Gene expression was quantified by BioClavis using the TempO-Seq Human Whole Transcriptome v2.0 assay, which targets 22,537 probes corresponding to 19,687 genes [63]. Data were provided by BioClavis as raw probe-level count tables.

For biomarker analysis, raw probe counts were first transformed to the gene level. Counts were normalized to counts per million (CPM), multiplied by the corresponding attenuation factor, and averaged across probes mapping to the same gene. The resulting gene-by-sample matrix was log_2_-transformed and used for principal component analysis (PCA) with the stats R package (v4.4.2), as well as for single-sample gene set enrichment analysis (ssGSEA) [64] implemented in the GSVA package (v2.07). This approach generated normalized enrichment scores (NES) for 866 gene sets of curated markers for human cell types derived from single-cell sequencing studies. Gene sets were obtained from MSigDB C8 (v2025.1, accessed on 8 January 2025). NES values were subsequently averaged within each experimental group.

For analysis of gene expression changes, data processing followed the recommendations of the R-ODAF guidelines for regulatory omics applications [65]. Only probes detected in at least 75% of samples within one experimental group were retained, applying a detection threshold of 1 CPM. To obtain a gene level expression matrix, individual probe counts were summed for each gene. Samples were excluded if their sequencing depth was below 1 × 10^6^ reads, corresponding to one third of the targeted depth of 3 × 10^6^ reads. Furthermore, samples with low correlation to the group mean (CPM-normalized counts), outside of Tukey’s Outer Fence (3× interquartile range; Tukey, 1977), were excluded. Additional sample quality control parameters were assessed, similar to Harrill et al. [66] (Appendix A).

To visualize transcriptional responses across tested conditions, dimensionality reduction was performed with uniform manifold approximation and projection (UMAP) [67], implemented in the umap R package (v0.2.10), using log_2_ CPM-normalized counts as input.

For differential expression analysis DESeq2 (v1.46) [68] was used. Log_2_ fold change was estimated using shrinkage (“normal”). A gene was considered if the differentially expressed if the absolute log_2_ fold change was at least 1 and the adjusted p value was below 0.01 (Wald test with Benjamini–Hochberg correction). Overlap between differentially expressed gene sets was calculated with the GSVA package (v2.07) as the proportion of overlapping genes relative to the smaller gene set.

To capture gene expression change patterns beyond single-gene changes, co-expression network analysis was performed using WGCNA (weighted gene correlation network analysis; WGCNA R package v1.73) [69]. Networks were constructed using bicor correlation in an unsigned network, with a soft power threshold of 16 chosen based on the scale-free topology criterion. Modules were derived with the parameters minModuleSize = 10, deepSplit = 3, and pamRespectsDendro = FALSE, yielding 50 modules. For each treatment group, eigengene scores (EGs) were calculated for every module. To obtain these scores, the log_2_ fold changes of all genes within a module were summarized using principal component analysis, with the first principal component taken as the EG. These module-level profiles were then functionally interpreted. Modules were annotated using overrepresentation analysis (ORA) with enrichR (v3.4). Pathway enrichment was assessed against KEGG (Human), Reactome, BioPlanet, and Gene Ontology Biological Process databases, all accessed on 7 April 2025.

Furthermore, to identify potential upstream regulators of the signaling pathways activated by the different stimuli, transcription factor (TF) activity was estimated using the log_2_ fold-change matrix. The analysis was carried out with the decoupleR package (v2.7) [70].

### 2.13. Immunofluorescence Staining

MGLCs were seeded either on 1 µg/mL fibronectin in DPBS-coated glass cover slips (Thermo Fisher Scientific, Waltham, MA, USA) or 96-well plates at a density of 50,000 cells/cm^2^ in respective media. Cells were fixed by replacing the medium with 10% neutral buffered formalin (Leica Biosystems Richmond, Inc, Richmond, IL, USA) in DPBS and incubated at room temperature for 10 min. Fixation solution was removed, and cells were washed once with DPBS. Cells were permeabilized in 0.6% Triton-X100 (Sigma-Aldrich, Traufkirchen, Germany) in DPBS at room temperature for 10 min. Afterwards, cells were blocked with blocking buffer (0.1% Triton-X100 and 5% FBS in DPBS) at room temperature for 1 h. Incubation with the respective primary antibodies (Appendix A) at indicated dilutions in blocking buffer was performed at 4 °C overnight. Non-bound primary antibodies were removed, and cells were washed three times with DPBS. Then, incubation with the respective secondary antibodies (Appendix A), diluted in blocking buffer, was performed for 30-60 min at room temperature. Nuclei were counterstained by the addition of Hoechst-33342 together with the secondary antibodies. After washing three times with DPBS, cells in 96-well plates were stored in DPBS at 4 °C until further use, and cells grown on glass cover slips were mounted using Aqua-Poly/Mount (Polyscience, Warrington, PA, USA) face-down onto object slides, left to dry overnight at room temperature, and then stored at 4 °C until further use. Cells were imaged by confocal microscopy (Zeiss LSM 770, 63x/1.4 Plan Apochromat Oil objective; Zeiss, Oberkochen, Germany), and images were processed in ImageJ (FIJI version 1.54f).

### 2.14. NFκB Translocation

NFκB translocation was measured in MGLCs and astrocytes. MGLCs on DoC4 were seeded on fibronectin-coated 96-well plates at a density of 50,000 cells/cm^2^ in MGLC differentiation medium, and experiments were conducted with DoC7 MGLCs. Astrocytes were seeded on poly-L-ornithine/fibronectin/laminin-coated (as described above) 96-well plates at a density of 40,000 cells/cm^2^ in astrocyte differentiation medium, and experiments were conducted after two to three days of reseeding.

Cells were stimulated with different inflammogens (see inflammogens) or medium only (control) and incubated at 37 °C and 5% CO_2_ for 1 h. Afterwards, cells were fixed by replacing the medium with 10% neutral buffered formalin (Leica Biosystems Richmond, Inc, Richmond, IL, USA) and incubated for 10 min at room temperature. Fixation solution was removed, and cells were stored in DPBS at 4 °C. Immunofluorescence staining was performed as described using anti-NFκB p65 antibody (Appendix A), and nuclei were counterstained with Hoechst-33342 (H-33342; Merck, Darmstadt, Germany).

The nuclear factor B (NFκB) translocation in MGLCs was quantified as follows: The NFκB p65 subunit was quantified using a Cellomics ArrayScan automated microscope with the predefined algorithm “nuclear translocation” as described previously [17,57,71]. In brief, the outline of the nuclei in the H-33342 channel was determined first. From there, the mean average pixel intensities of the nuclear area (measured 3.3 µm = 5 pixels from the nuclear outline towards the nuclear center) and the mean average pixel intensities of a ring in the cytoplasm (width of 2.6 µm = 4 pixels and 3.3 µm = 5 pixels away from the nuclear outline towards the cell membrane) were measured. The ratio of the average pixel intensities of the “nuclear area” and the “cytoplasmic area” were calculated. “Activated cells” were defined when the ratio exceeded the average ratio of unstimulated reference cells by at least one SD. The average intensity ratio of the reference wells was automatically obtained. In each well, 300 cells were automatically used for analysis. For analysis, at least three independent MGLC differentiations were used, and images were taken from four technical replicate wells each.

NFκB translocation in astrocytes was quantified as follows: The translocation of the NFκB p65 subunit was quantified using an automated ImageXpress Nano Automated Cell Imaging System (Molecular devices, Sunnyvale, CA, USA) by using the “Enhanced-Translocation” module of MetaXpress (Molecular devices, Sunnyvale, CA, USA). Images were acquired at 3 random sites of each well, capturing roughly 300–500 cells per site (900–1500 cells per well). Briefly, the algorithm used the H-33342 staining to automatically identify the outline of the nuclei. The nuclear area was defined as the inner area within the nucleus that is located more than 1 µm inward from the nuclear outline (inner region distance in from edge). The cytoplasmic area of each cell was defined as the region extending outward from the nucleus, starting 1 µm beyond the nuclear outline (outer region distance out from edge), forming a ring-shaped region around the nucleus with a width of 3 µm (outer region width). For each cell, the ratio of p65 intensity in the nucleus and in the cytoplasmic ring defined as the Nuc/Cyt p65 NF-κB ratio was calculated. Cells were scored as positive for nuclear translocation of p65 if the correlation coefficient was 0.65 or greater. Further input parameters were entered as follows: approximate maximum width = 15 μm^2^; intensity above background = 200; maximum intensity above background = 10,000; minimum area = 30 μm^2^; and maximum area = 300 μm^2^. For analysis, at least three independent astrocyte differentiations were used, and images were taken from four technical replicate wells each.

### 2.15. Supernatant Transfer and TNFα Neutralization

An anti-TNFα monoclonal antibody Adalimumab/Humira (AbbVie Biotechnology GmBH, North Chicago, IL, USA) was used to neutralize TNFα in medium. The neutralizing capacity of the antibody was determined with serial dilutions of anti-TNFα antibody (starting concentration: 10 µg/mL) with 20 ng/mL TNF for 2–3 h at 37 °C. To neutralize TNFα released from stimulated MGLCs into the supernatant, the medium was incubated with 100 ng/mL of Adalimumab at 37 °C and 5% CO_2_ for 2–3 h. Supernatants without the addition of Adalimumab were kept under the same conditions as neutralized samples.

### 2.16. Western Blot

For Western blot analysis of proteins, 1 Mio DoC4 MGLCs were seeded into each well of a 6-well plate (Corning, New York, NY, USA). After three days, 20 µM ruxolitinib (Selleckchem, Munich, Germany) was added to some wells. After 30 min, some of the inflammatory stimuli (see above) were added and incubated for 1 h. Then, cells were lysed in 115 µL 1× Laemmli buffer and heated for 5 min to 95 °C. Lysates were centrifuged for 1 min at 10,000× *g* through NucleoSpin Filters (Macherey-Nagel GmbH, Düren, Germany) to break down long DNA strands. 20 µL of lysates were loaded onto 10% SDS gels, and gels were run for 30 min at 80 V and then at 120 V until bromophenol blue bands reached the bottom of the gel. Proteins were transferred onto nitrocellulose membranes (Amersham, Buckinghamshire, UK) using the iBlot 2 dry blotting system (Invitrogen, Waltham, MA, USA). Membranes were blocked with 5% (*w*/*v*) bovine serum albumin in (TBS)-Tween (0.5% (*v*/*v*)) (both from Roth, Karlsruhe, Germany) overnight. Respective primary antibodies were incubated at 4 °C overnight (Appendix A). Membranes were washed three times in TBS-Tween (0.5% (*v*/*v*)) at room temperature for 10 min and then incubated with horseradish peroxidase-conjugated secondary antibodies for 1 h at room temperature (Appendix A). ECL Western blotting substrate (Pierce/Thermo Fisher Scientific, Rockford, IL, USA) was used for visualization, and chemiluminesence was recorded with a Fusion-SL 3500 WL device with Fusion software (Version 15.18; Bio-Rad, Hercules, CA, USA). For densitometric quantification, in-house software was used. The integrated intensity of pixels of a given protein was quantified. Background correction was performed by a single region with an unspecific background signal. A list of used antibodies can be found in the Appendix A.

### 2.17. Statistical Analysis

Unless stated otherwise, displayed data are the means (±SD or ±SEM) of three biological replicates (different cell preparations). Biological replicates are indicated by “N”, and technical replicates are indicated by “n” in the figures or figure legends. If not otherwise indicated, one technical replicate was performed per biological replicate. Each biological replicate was calculated as the mean of at least three to five technical replicates (different wells within the same experiment). The sample size of technical replicates was chosen based on intra-experimental variance. Information concerning descriptive statistics and experimental variability is included in the figure legends or the figures themselves. For data visualization, GraphPad Prism (V8.0.2; GraphPad Software, Inc., San Diego, CA, USA) was used. Raw data have been deposited on Zenodo and are publicly accessible under the following DOI: 10.5281/zenodo.17199104 [72].

## 3. Results and Discussion

### 3.1. Characterization of Human Microglia-like Cells and Their Responses

To generate microglia-like cells (MGLCs), we adapted a two-step differentiation protocol from previously established methods [51,55,56,58]. Human induced pluripotent stem cells (iPSCs) were used to generate embryoid bodies (EBs), which were plated into cell culture flasks. The adherent EBs further differentiated into so-called myeloid factories, which produced non-adherent macrophage precursor cells (PreMacs). In a last step, PreMacs were further differentiated into MGLCs for about seven days (Figure 1A) until they displayed characteristic microglia morphologies with multiple fine processes (Figure 1B) [9,10,11]. MGLCs were positively immunostained for the canonical markers IBA-1, PU.1, TMEM-119, and S100β (Appendix A) [73]. To ensure robust and reproducible differentiation, the myeloid markers CD11b, CD14, CD16, and CD68 were regularly measured by flow cytometry (Figure 1C and Appendix A). Both PreMacs and day of culture (DoC) 7 MGLCs consistently expressed these markers in ≥80% of cells throughout the differentiation (Figure 1C,D, Appendix A).

Transcriptome analysis revealed the expected gene expression profile changes during differentiation from PreMacs to DoC1 and DoC7 MGLCs (Appendix A). A principal component analysis (PCA) of the gene expression patterns of the factory cells (d45), PreMacs, and MGLCs on DoC1 and DoC7 showed that each cell type had a distinct profile (Figure 1E). To further evaluate lineage identity, we selected markers typically expressed in microglia, macrophages, lymphoid cells, astrocytes, and neurons. Notably, microglia-associated genes such as *AIF1* (encoding IBA-1), *CX3CR1* (the IL-34 receptor critical for microglial survival), and *P2RY12* (a microglia-specific marker) were upregulated in DoC7 MGLCs (Figure 1G and Appendix A). Additional microglial identity markers [51,56], including *MERTK*, and *OLFM3* were upregulated over time. The results of single sample gene set enrichment analyses (ssGSEA) showed that genes related to a macrophage/microglia identity were selectively upregulated. For comparison, gene sets related to hepatocytes (other organ) or astrocytes (other brain cell type) were examined in similar ways and found not to be enriched (Figure 1F).

Given that microglia respond to inflammatory stimuli with morphological and functional changes, MGLCs were stimulated for 48 h with ligands targeting different Toll-like receptors (TLRs) or with TNFα (TNF). Exposure to the TLR4 ligand lipopolysaccharide (LPS) induced a star-shaped/activated morphology, whereas stimulation with the TLR3 ligand poly(I:C) (I:C) resulted in an elongated, polarized cell morphology (Figure 1H). Stimulation with the TLR2/6 ligand FSL-1 (FSL) or with TNF resulted in morphologies similar to LPS-treated cells. Other tested stimuli such as the TLR7 ligand imiquimod (IQM), interferon-γ (IFN), or exposure to dexamethasone (DEX) did not clearly affect cell morphology compared to untreated control cells (Appendix A). The different cell morphologies observed suggest that MGLCs are not clearly in a resting or activated state, but rather different types of activation states may be assumed.

To test this hypothesis on a functional level, MGLCs were exposed to LPS or I:C. The secretion of IL-6, IL-8, and IL-1β was measured at 4, 8, and 18 h after stimulation. Both stimuli resulted in similar IL-6 secretion (Figure 1I). However, LPS triggered a more pronounced IL-8 secretion than I:C, while the opposite response was observed for IL-1β (Figure 1I). These findings suggest that different types of inflammatory responses can be triggered, and that the stimulus used is an important determinant. This ability to respond stimulus-specifically has been demonstrated before in a study on murine microglia responses to LPS and I:C. Different cytokine release profiles were detected [74]. Similarly, human-monocyte-derived macrophages exhibited distinct cytokine secretion in response to different TLR stimuli (including LPS and I:C) [75], and human iPSC-derived astrocytes displayed markedly different transcriptomic patterns following TNF or IFN stimulation [17]. The diversity of activation states of inflammatory cell types has been discussed earlier [14,15,16], and our initial data indicated that this deserves further exploration for human MGLCs. The distinct, stimulus-dependent responses were therefore further characterized in subsequent experiments.

### 3.2. Inflammation-Induced Transcriptome Changes in Microglia-like Cells

To obtain a comprehensive overview of MGLC responses after inflammatory stimulation, transcriptome analyses were performed. Cells were exposed to a panel of stimuli for 4, 8, or 18 h (Figure 2A, complete data can be found in the Appendix A). At first, we aimed to determine the expression levels of receptors targeted by the respective stimuli used to trigger the inflammatory responses. Transcripts for *TLR1-8* (except *TLR2*) as well as for *TNF* (*TNFRSF1A*, *TNFRSF1B*), *IFN* (*IFNGR1*, *IFNGR2*), and glucocorticoid receptors (*NR3C1*) were detected (Figure 2B). This analysis exemplifies how focused subsets of our transcriptome data may be used in the future, e.g., to map signaling pathways or to gain an overview of expressed adhesion molecules. To obtain an overall picture of transcriptome changes, we used uniform manifold approximation and projection for dimension reduction (UMAP) analysis. This approach provided information of high granularity and a largely better separation of the effects that were observed in a PCA. Clear stimulus- and time-dependent patterns, with partial but not complete segregation between treatments were observed (Figure 2C). This is in line with the known advantages of UMAP for analysis of high-dimensional data sets that contain both linear and non-linear responses (typical for transcriptome data) [67,76,77,78]. One clear outcome was the shared transcriptional responses, e.g., between LPS and FSL. The UMAP data also highlighted major differences, e.g., between LPS and IFN.

The analysis of differentially expressed genes (DEGs) demonstrated a generally strong responsiveness of MGLCs to all stimuli (Figure 2D and Appendix A). The fewest DEGs were induced by IQM after 4 h (225 up- and 50 downregulated genes; at 18 h: about 1900 up and 1700 down), whereas LPS at 4 h triggered the largest changes (about 2700 up- and 2500 downregulated genes). Also, I:C evoked a strong overall transcriptional response, consistent with previous reports on a potent I:C-mediated activation of microglia [74,79,80]. The DEG counts peaked at different time points, depending on the stimulus: e.g., at 4 h for LPS, 8 h for I:C and IFN, and 18 h for IQM and TNF. These findings indicate that data for several time points are required for capturing the different, highly dynamic inflammatory processes. Notably, the co-treatment of LPS with the glucocorticoid receptor against DEX did not markedly reduce DEG numbers relative to the LPS response. However, when comparing the total number of fold changes per condition, a decrease from LPS to LPS + DEX was observed (Appendix A). This decrease in effect size suggests that DEX tended to attenuate the extent of gene regulation rather than to abrogate LPS-induced transcript changes completely. This observation is in line with the complex roles of the glucocorticoid receptor in tissue microglia, where responses range from inflammatory attenuation to priming, depending on the timing of receptor activation [81,82].

Next, we provide an overview of the degree of similarity between the 24 types of inflammatory responses observed here. For this, we calculated the pairwise overlap ratio (proportion of genes shared between two conditions). TLR ligand treatments showed a substantial overlap with TNF, whereas IFN and DEX were transcriptionally distinct from, e.g., LPS or I:C responses as well from one another. The response to LPS plus DEX was similar to the DEX response but also shared genes with the LPS-response (Figure 2E).

Weighted gene co-expression network analysis (WGCNA) was used to identify groups of genes (modules) that were regulated in the same way across the different conditions. All available data sets were used to construct a co-expression network and to identify modules of correlated genes (Appendix A).

A subset of the modules was selected for further analysis based on their relevance to microglia activation (Figure 2F,G). A comparative overview of the six selected modules activated by the different stimuli was obtained. “TNF signaling” was consistently upregulated following stimulation with FSL, LPS, and I:C (module #6). “Oxidative stress response” modules were upregulated for the TLR ligands as well as for TNF, which indicates oxidative stress as a downstream consequence of inflammatory activation (modules #32, and #13). MHC class II protein complex assembly seemed to be especially upregulated after 18 h, suggesting a transition toward antigen presentation and potential crosstalk of the innate immune response with the adaptive immunity (module #15). Also, a broad activation of (immune) defense responses was observed, especially for LPS and I:C (module #24). Furthermore, modules linked to antiviral and interferon-alpha (IFNα) responses were upregulated after IFN and I:C treatment. Similarly, though less pronounced, these modules were also enriched following LPS and FSL stimulation, likely due to shared gene expression patterns (module #12).

Besides this overarching analysis based on modules, several single genes were also investigated to identify stimulus-specific marker genes of the responses. We identified *CXCR4* (a key regulator of immune cell migration) as being exclusively upregulated by IQM, particularly at the later time points (8 and 18 h) (Appendix A). *IL31RA* expression was elevated in response to IFN and I:C and may serve as biomarker for IFN-type responses. *MT1G*, an Nrf2 target with anti-oxidant and neuroprotective properties [83,84,85], was upregulated following exposure to all TLR ligands and TNF at 18 h. It may serve as indicator of typical LPS-type responses. To identify potential upstream regulators of the signaling pathways triggered by the different stimuli, the overrepresentation of transcription factor (TF) binding sites was determined in the gene sets altered by inflammatory stimuli. A clear overrepresentation of TF binding sites for NFκB was revealed in the set of transcripts triggered by TLR ligands and TNF. Binding sites for the JAK/STAT signaling pathway were overrepresented amongst the genes triggered by IFN (Appendix A). Our analyses of the inflammatory transcriptome responses addressed different levels of organizational complexity, ranging from individual genes, over biologically and statistically linked gene sets towards measures on the level of the whole transcriptome. All of them pointed to a clear diversity of MGLC responses, even though there was an overlap for some (related) stimuli.

### 3.3. Divergent Signal Transduction in MGLCs After Exposure to Inflammatory Stimuli

NFκB translocation is a signal transduction mechanism shared by many inflammogens (and their cognate receptors). To validate the results of the TF analysis, we quantified the fraction of cells exhibiting NFκB p65 translocation from the cytoplasm to the nucleus (Figure 3A,B). In control cells, NFκB p65 immunoreactivity was relatively homogeneously distributed throughout the cell (often with stronger cytoplasmic than nuclear signals). Upon treatment with LPS or I:C, NFκB p65 translocated from the cytoplasm into the nucleus within 1 h. This led to a distinctly higher signal in the nuclear area than in the surroundings (Figure 3A and Appendix A). Cell viability was unaffected by any treatment for up to 48 h (Appendix A). Quantitative analysis of the imaging data revealed that over 90% of MGLCs exposed to LPS or I:C showed NFκB translocation and were classified as translocation-positive (Figure 3C and Appendix A).

These initial experiments showed that classification of cells as having translocated NFκB or not may be used as one of the read-outs for the inflammatory activation of MGLCs. This is in line with the successful use of this endpoint in human astrocytes [17]. Considering the scientific basis of the translocation assay, it is important to note that in some cells (e.g., astrocytes), NFκB is virtually absent from the nucleus in the non-stimulated state. In most other cell types, there is a fraction of NFκB (10–30%) already in the nucleus of resting cells, and this fraction increases with stimulation [86,87]. This is also observed for other transcription factors (e.g., phosphorylated c-Jun in microglia) [88] and was taken into account for our classification algorithm and the translocation assay. IFN treatment did not trigger any NFκB translocation, which is in line with the known signal transduction of IFN [31,32,33,34,35]. Treatment with FSL, IQM, and TNF induced nuclear translocation in approximately 60% of the cell population (Figure 3C and Appendix A). This may be due to only a subpopulation expressing the respective receptors. Another explanation could be that not all cells in the population allowed an efficient coupling of these receptors to the NFκB pathway. These results demonstrate that MGLCs are capable of responding to diverse inflammogens through the NFκB pathway although with different efficacy. The activation of subpopulations can also be seen in macrophage populations, where a few rare cells display different responses that help balance defense activation against pathogens with the aim to minimize self-harm (by avoidance of overstimulation) [75,89].

IFN signals via a fundamentally different route, activating the JAK-STAT pathway rather than NFκB [31,32,33,34,35]. To validate this in MGLCs, we assessed phosphorylation of STAT1 and STAT3 (P-STAT1, P-STAT3) (Figure 3D). Neither P-STAT1 nor P-STAT3 was detectable in control cells (Figure 3E,F and Appendix A). Stimulation with IFN alone or in combination with LPS induced robust phosphorylation of STAT1 and STAT3, with LPS co-stimulation showing no additive effect. Importantly, treatment with the JAK inhibitor ruxolitinib completely abrogated IFN-induced phosphorylation of STAT1 and STAT3, confirming specific JAK-STAT pathway activation by IFN. Thus, IFN activated the JAK-STAT axis in MGLCs, similar to its role in several other cell types. For the study of the effects of longer exposures to inflammatory stimuli, it needs to be considered that there may be crosstalk between the NFκB and JAK-STAT pathway. For instance, phosphorylation of STAT has been observed after 3 h stimulation with LPS or I:C stimulation [74]. Such findings may explain why the TF overrepresentation analysis also revealed some JAK-STAT pathway members (Appendix A). The crosstalk may occur intracellularly or more indirectly, e.g., by secretion of ligands that trigger additional receptors.

Altogether, the signaling experiments show that the iPSC-derived MGLCs react to diverse inflammatory stimuli in specific ways consistent with literature knowledge [90,91]. At least two clearly different pathways, NFκB and JAK-STAT, have been shown here to be engaged, and the TF overrepresentation analysis suggests several more to act in parallel. Further investigations are needed to elucidate details on how differential signaling events translate into stimulus-specific cytokine release profiles and a potentially different susceptibility to pharmacological interventions.

### 3.4. Release of Inflammatory Mediators from Microglia-like Cells

Cytokines and chemokines produced by microglia are central to regulating and fine-tuning CNS inflammation. We assessed the time-dependent release of mediators from MGLCs after exposure to a panel of inflammatory stimuli. A basal release (from resting cells) was found for MIP-1β, IL-8, IFN, IL-1Ra, IL-10, and CCL-2 (Figure 4A–C). This essentially aligns with previous findings reported by others [51,52,53]. The exposure to inflammatory stimuli led to an overall increase in cytokine and chemokine secretion: LPS and I:C triggered the strongest responses. The two TLR ligands mostly triggered similar release kinetics, and the concentrations of released mediators were within a similar range (±3-fold), except for IL-8 and IL-10. IQM produced the lowest cytokine levels among the stimuli. However, these levels consistently remained above control values except for five of the measured mediators. Early peaks in IL-1Ra, IL-10, and CCL-2 were seen with most stimuli but not with IQM (Figure 4C).

Stimulation with TNF produced noticeably lower cytokine responses compared to LPS (Figure 4D). This observation is consistent with findings by Knappe et al. [92]. Both their data and ours showed upregulation of CCL-2 and IL-8 among the measured cytokines. A potential physiological rationale for the weak pro-inflammatory response to TNF is that microglia are potent producers of TNF (when, e.g., stimulated by LPS or I:C), and they are considered the main source of TNF in the brain. The relatively low capacity of TNF to trigger cytokine release from microglia may prevent overstimulation, by avoiding a TNF-driven positive feedback loop. Instead, TNF from microglia may act primarily on astrocytes to activate this additional cell type to participate in the immune defense [39,41]. Relative to LPS, TNF induced more anti-inflammatory cytokines (IL-4 and IL-1Ra) than pro-inflammatory ones (Figure 4D).

IFN exposure did not induce significant changes in the measured cytokine or chemokine concentrations as compared to control conditions (indicating only a minimal direct effect on the cytokine/chemokine release under these conditions). This is in line with a large body of work that gives evidence on a primary role of IFN as enhancer of other microglia/macrophage stimuli when, e.g., present together with LPS [93,94,95,96].

### 3.5. Differential Astrocyte Activation by Microglia-like Cells Exposed to Different Inflammogens

Glial cells within the CNS engage in complex interactions [36,37,38,39]. This prompted us to investigate how different inflammatory states of MGLCs may affect astrocytes. Previously, it has been shown that translocation of NFκB p65 into the nucleus may be used as readout for astrocyte activation [17,57,97,98,99]. Consistent with this, we observed a concentration-dependent NFκB p65 nuclear translocation in astrocytes after TNF treatment (Figure 5A) and decided to use this system to characterize the functional consequence of MGLC stimulations. First, we directly stimulated astrocytes with the different inflammogens to assess their responsiveness to stimuli that may be carried over from activated microglia cultures. LPS, FSL, IQM, and IFN stimulation did not trigger NFκB translocation above baseline (Figure 5B,C and Appendix A). NFκB translocation as a response to I:C was observed in approximately 40% of astrocytes (Figure 5B and Appendix A; Quantification of the medium control can be found in Appendix A). The data suggest that functional TLR4 signaling is not present in astrocytes and that TLR3 is only expressed in a subpopulation. This is in agreement with reported astrocyte TLR3 profiles and an absence of LPS responses [17,40,97,100,101,102,103]. To evaluate astrocyte activation by microglia, we treated MGLCs for 18 h with inflammogens, transferred the conditioned supernatant (SN) to astrocytes, and measured NFκB translocation in these cultures (Figure 5D). Astrocytes treated with either medium or medium from unstimulated microglia led in less than 10% of astrocytes to NFκB translocation. Several SNs, including those from LPS- and FSL-stimulated MGLCs, induced NFκB translocation in astrocytes (Figure 5E–G). This suggests that MGLCs secrete factors capable of astrocyte activation. Since TNF is a key microglia-derived cytokine that activates astrocytes [39,41], the TNF release from MGLCs after 18 h of stimulation was quantified. Indeed, sufficient TNF was secreted by LPS-, I:C- and FSL-stimulated MGLCs to drive astrocyte activation (Figure 5H). To confirm that TNF is the primary signaling cytokine mediating astrocyte activation, the aim was to neutralize TNF in the conditioned medium. The required antibody concentration was determined (Appendix A). Application of a sufficient but not overwhelming concentration of anti-TNF antibodies completely blocked the astrocyte NFκB response to, e.g., LPS SN (Figure 5I,J). This clearly shows that TNF plays a central role in transmitting some microglia inflammatory responses to astrocytes. Essentially, similar findings were obtained for SN from MGLCs stimulated with FSL. Also, under this condition astrocyte activation that could be inhibited by anti-TNF was observed.

The interpretation of other stimulation conditions was more complex. I:C directly induced NFκB translocation in some astrocytes (40%), while SN from I:C-stimulated MGLCs triggered a stronger response (80% of cells) (Figure 5B,I). To exclude that anti-TNF may have interacted with I:C, I:C was re-added to astrocytes exposed to I:C SN in the presence of anti-TNF. Under this condition, NFκB translocation occurred in 40% of the cells (Figure 5I). The most likely explanation is that MGLCs internalize I:C (note that it acts via intracellular receptors, Figure 2A) so that no I:C is left after SN transfer to directly activate astrocytes. This suggests that TNF triggered by I:C in MGLCs is present in transferred SN and triggers NFκB translocation in astrocytes. A complete block of astrocyte activation by anti-TNF supports this assumption.

## 4. Conclusions

Here, we generated iPSC-derived microglia-like cells to characterize their response to a diverse panel of inflammatory stimuli, including four TLR ligands (LPS, I:C, FSL-1, IQM), TNF, and IFN. The cell responses were characterized at the transcriptional level by measuring cytokine release, by obtaining measures of intracellular signaling, and by considering consequences for the activation of astrocytes.

Several studies have addressed the inflammatory activation of human microglia and of related systems (like the MGLCs used here) [92,104,105,106]. These findings have provided an important basis for addressing more complex biomedical questions. Such future research directions comprise (i) the interaction of microglia with other brain cells, (ii) the use of microglia for the pharmacological and toxicological characterization of drugs and other chemicals, and (iii) the role of microglia in the human host defense. Our study represents one of the steps in these directions. For instance, we provided a comprehensive set of data on the activation of astrocytes by stimulated microglia. TNF was identified as one of the key mediators for some stimuli. In the future, additional stimuli, including complex ones, like microorganisms, may be used. Moreover, other potential intercellular messengers and their effects on astrocytes should be investigated. Analyzing changes in neurotrophic factor release and inflammatory cytokine secretion from astrocytes or quantifying gliosis markers like GFAP and vimentin would yield even more information on potentially different responses of astrocytes triggered by MGLCs, e.g., stimulated by LPS vs. I:C.

Another aspect of cell interactions concerns the impact of activated MGLCs on neuronal function and survival. This has been investigated in the past mainly in rodent systems [74,81,107,108,109,110] and may now be verified with human cells.

Our study also pointed out one approach, how functional MGLC responses may be used for compound screening. We showed that the NFκB translocation assay captures microglial responses to various inflammatory insults. This allows testing of whether different compounds modulate these responses similarly or differently. This approach could help to identify candidate compounds that modulate microglial activity, particularly those targeting unwanted or excessive inflammation. Targeted modulation of human microglia could open a therapeutic avenue for the treatment of a broad spectrum of neurodegenerative disorders [37,111]. Future studies using multiple donor-derived iPSC lines will enable evaluation of inter-individual variability and strengthen the translational generalizability of this platform.

Finally, a key strength of our work lies in its systematic approach: Previous studies have tended to examine a single time point, readout, or stimulus, while we assessed multiple layers of microglial biology (transcriptome, cytokine release, nuclear translocation of NFκB, and cell morphology) in response to a broad set of inflammogens. This provides a more comprehensive picture of the diversity of acute host defense responses to be expected in the human brain. For instance, studies investigating only a single time point (typically 24 h post-stimulation) may overlook fast, transient, or evolving inflammatory dynamics. Our data on early and mid-phase responses capture the full dynamic spectrum of human microglial activation. While we are aware that only a small subset of inflammatory situations has been covered to date, our study provides extensive information on a robust, relatively high-throughput model system to be used for further studies.

## Figures and Tables

**Figure 1 cells-14-01687-f001:**
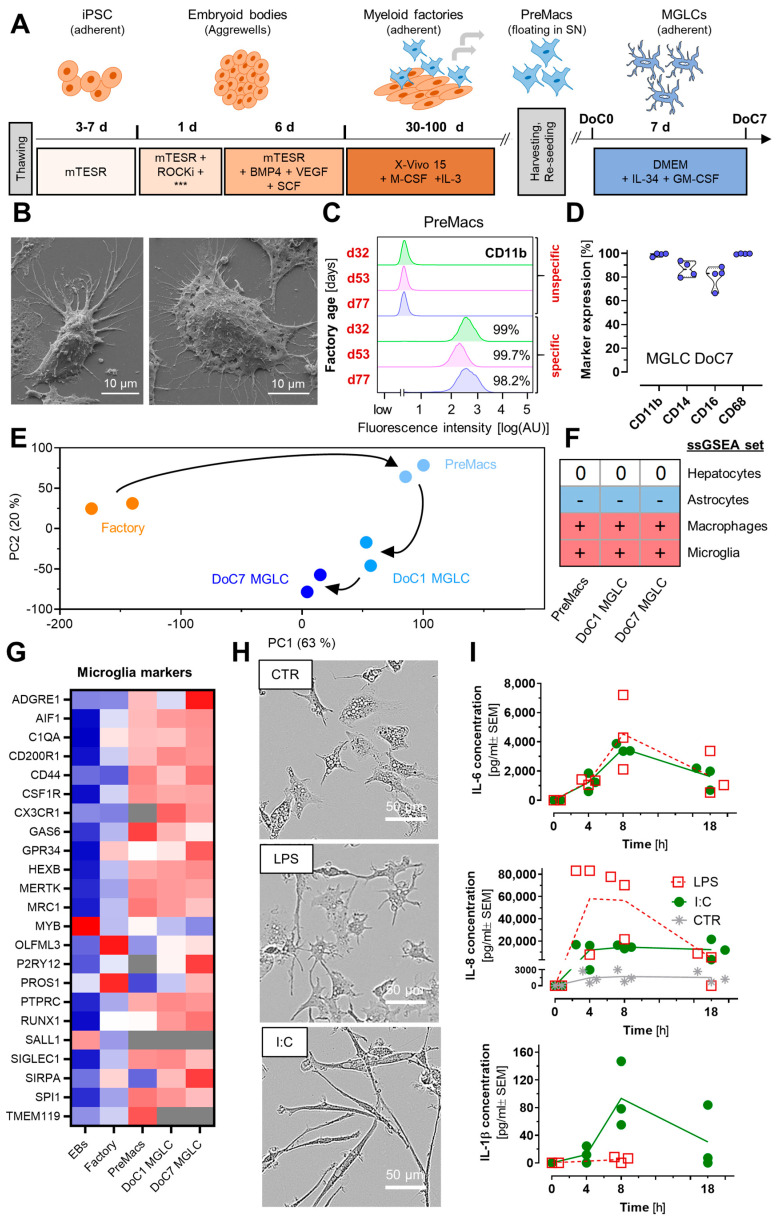
Characterization of human microglia-like cells and their responses. (**A**) Schematic diagram of the differentiation protocol of microglia-like cells (MGLCs). Media, including growth factors (*** = BMP4, VEGF, SCF) and differentiation times, are depicted. Human induced pluripotent stem cells (iPSCs) were used to form embryoid bodies (EBs). These were plated into cell culture flasks to generate myeloid factories in the presence of M-CSF and IL-3. Non-adherent macrophage precursor cells (PreMacs) were harvested from the supernatant medium and further differentiated to MGLCs with IL-34 and GM-CSF. The time axis for the myeloid factories is given in days (d). For the MGLCs, days of culture (DoC) are counted after their harvest from the myeloid factories and their subsequent plating. (**B**) Representative scanning electron microscopy (SEM) images of DoC7 MGLC phenotypes after 1 h of attachment. (**C**) Flow cytometry measurement of CD11b in PreMacs, directly after harvest from the myeloid factories on d32, 53, and 77 (see more detailed data in Appendix A). Samples were labeled with a CD11b antibody (specific) or not (unspecific). The percentage of CD11b-positive cells is indicated. (**D**) Flow cytometric quantification of CD11b, CD14, CD16, and CD68 in MGLCs on DoC7 (N = 4). (**E**) The gene expression of myeloid factories (d45), PreMacs, and MGLCs on DoC1 and DoC7 was quantified by transcriptome analysis. In the 2-dimensional display of the principal component analysis (PCA), the axes are scaled according to the variance explained by the respective PCA component. Differentiation stages are color-coded. Arrows indicate the rough trajectory within the 2D PCA space (N = 2). (**F**) Average single sample gene set enrichment analyses (ssGSEA) scores for hepatocyte-, astrocyte-, macrophage-, and microglia-related gene sets are shown for PreMacs and MGLCs on DoC1 and DoC7. A ssGSEA score of <−1000 is indicated as “-”, a score of 0–1000 as “0” and a score of >3000 as “+”. See Appendix A for raw data (N = 2). (**G**) Heat map depicting the row-wise z-scores of gene expression changes of selected marker genes in EBs, myeloid factories (d45), PreMacs and MGLCs on DoC1 and DoC7. Fully saturated blue indicates a z-score of −1. Fully saturated red indicates a z-score of +1. Less saturated colors indicate intermediate values. Pure white indicates a z-score of zero. Gray fields indicate samples with a signal below the detection limit (N = 2). (**H**) Exemplary bright field images of MGLCs on DoC9 treated for 48 h with medium (CTR), LPS (100 ng/mL), or I:C (10 µg/mL). (**I**) IL-6, IL-8, and IL-1β were measured in cell supernatants at 4 h, 8 h, and 18 h after stimulation of DoC7 MGLCs with LPS (100 ng/mL) or I:C (10 µg/mL). No data are shown when the concentration was below the lower limit of quantification (LLOQ) of 13 pg/mL (IL-6), 38 pg/mL (IL-8), and 16 pg/mL (IL-1β) for all data points under a certain condition (time point and stimulus). If the concentration was not below the LLOQ for all data under a certain condition (time point and stimulus), data below the LLOQ were assumed to be zero. The concentration of IL-8 in CTR at ≥4 h was above the upper limit of quantification of 83,160 pg/mL. Therefore, these data points are shown. Data are means of the biological replicates ± SEM (N = 3).

**Figure 2 cells-14-01687-f002:**
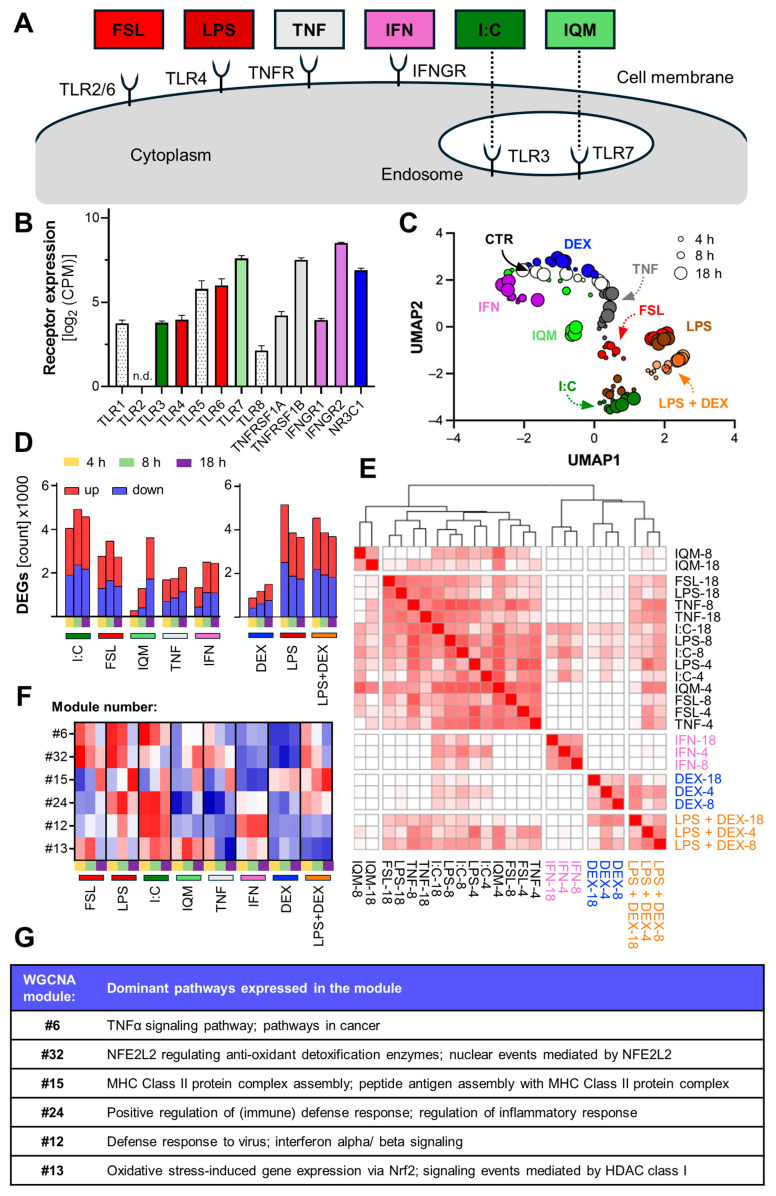
Inflammation-induced transcriptome changes in microglia-like cells. (**A**) Schematic illustration of toll-like receptors (TLRs) and their ligands: FSL-1 (FSL), lipopolysaccharide (LPS), Poly I:C (I:C), and imiquimod (IQM). Also displayed are TNFα (TNF), interferon-γ (IFN), and their respective receptors. (**B**) On DoC7, MGLCs were treated with different immunogenic stimuli: LPS (100 ng/mL), I:C (10 µg/mL), FSL (100 ng/mL), IQM (10 µg/mL), TNF (10 ng/mL), IFN (20 ng/mL), DEX, or LPS *plus* DEX. After the respective treatment period (4 h, 8 h, or 18 h), cells were lysed, and gene expression profiling was performed using transcriptome analysis. All data in this figure were obtained using this workflow (N = 4). In unstimulated DoC7 MGLCs, gene expression of the receptors for TLR1-8 and the subunits of receptors of TNF, INF, and the glucocorticoid receptor was determined. The raw counts were normalized to counts per million total counts (CPM) and log_2_-transformed. n.d. = not detected. (**C**) The overall data structure of MGLC transcriptome is visualized using uniform manifold approximation and projection (UMAP) as a non-linear dimensionality reduction approach. Each point corresponds to one of four biological replicates per inflammatory stimulus (color-coded) and time point (size-coded). (**D**) The absolute numbers of differentially expressed genes (DEGs) per stimulus and time point are depicted. Up- and downregulated DEGs are represented as stacked bars in red and blue, respectively. Genes were considered as differentially expressed when the absolute log_2_ fold change was ≥ 1 and the adjusted *p*-value (padj) < 0.01. (**E**) Graphical representation of the correlation of gene expression across different conditions (as described in (**B**)). The overlap of all pairwise combinations of stimulation conditions was calculated as the ratio of shared genes between two exposure conditions to the number of DEGs in the smaller gene set. The degree of color saturation indicates the % correlation (white ≤ 40%, full red = 100%). DEGs were defined as in (**D**). (**F**) The overall set of transcriptome data was dimensionality-reduced (see Appendix A for raw data) by weighted gene co-expression network analysis (WGCNA). Using this approach, co-induced genes were grouped into modules. See Appendix A for a complete overview. Six modules were selected to show their activation (quantified by z-scores of the module eigengenes) by the various inflammatory conditions. The colors in the heat map represent the strength and direction of the module activation (red for positive and blue for negative gene regulation). (**G**) A short biological characterization of the modules shown in (**F**) is provided.

**Figure 3 cells-14-01687-f003:**
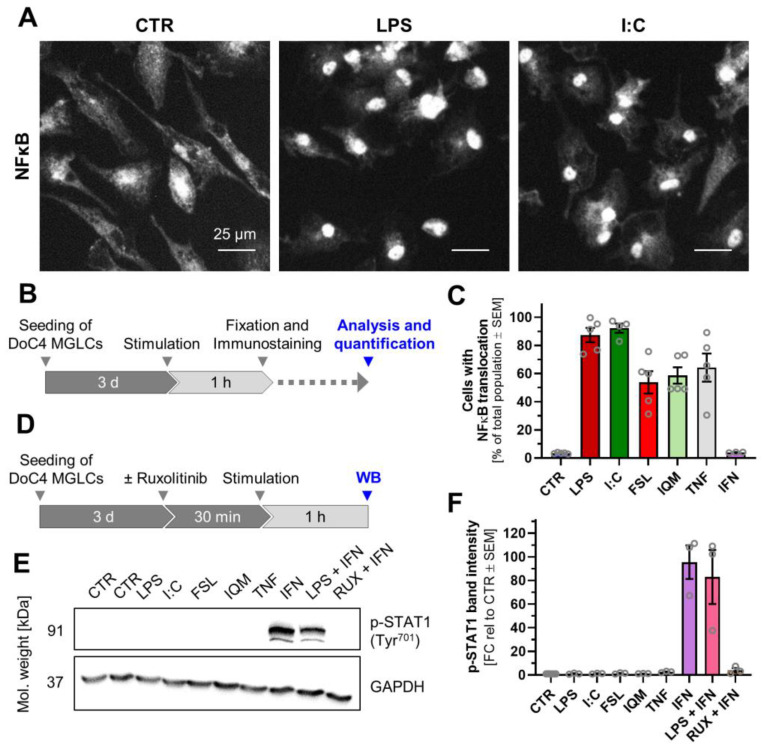
Divergent signal transduction in microglia-like cells after exposure to inflammatory stimuli. (**A**) Representative immunofluorescent images of MGLCs, showing the NFκB p65 signal after stimulation with LPS, I:C, or solvent (CTR). The nuclear counterstain (Hoechst-33342) is not shown here but is illustrated in Appendix A. (**B**) Schematic representation of the experimental protocol used for the determination of the NFκB translocation. On DoC7, MGLCs were treated with different stimuli: LPS (100 ng/mL), I:C (10 µg/mL), FSL (100 ng/mL), IQM (10 µg/mL), TNF (10 ng/mL), or IFN (20 ng/mL). After 60 min of treatment, MGLCs were fixed and immunostained against the NFκB subunit p65. (**C**) Imaging and quantification of NFκB translocation was performed on an automated high-content imaging device. Data are shown as the percentage of cells classified as having undergone NFκB translocation (from cytosol to the nucleus). Data are means of the biological replicates ± SEM (N = 3–5; n = 4). (**D**) Schematic representation of the experimental protocol used for Western blot analysis. MGLCs were treated with different stimuli for 1 h before sampling for Western blot analysis. MGLCs were optionally pre-treated with ruxolitinib (RUX, 20 µM) for 30 min. Data are means of three independent MGLC differentiations ± SEM. (**E**) Western blots were performed with MGLC lysates, using anti-phosphoSTAT1 (Tyr^701^) and anti-GAPDH antibodies. Besides the treatment mentioned in (**C**), the combinatory treatments LPS (100 ng/mL) *plus* IFN (20 ng/mL) and ruxolitinib (RUX, 20 µM) *plus* IFN (20 ng/mL) were used. The displayed blot is representative of three independent experiments. (**F**) Quantification of phosphoSTAT1 (Tyr^701^) Western blot bands. Bands were normalized to the intensity of the respective loading controls (GAPDH). Fold change is displayed relative to control. Data from all experiments are shown in Appendix A. Essentially similar data were obtained for STAT3 (N = 3).

**Figure 4 cells-14-01687-f004:**
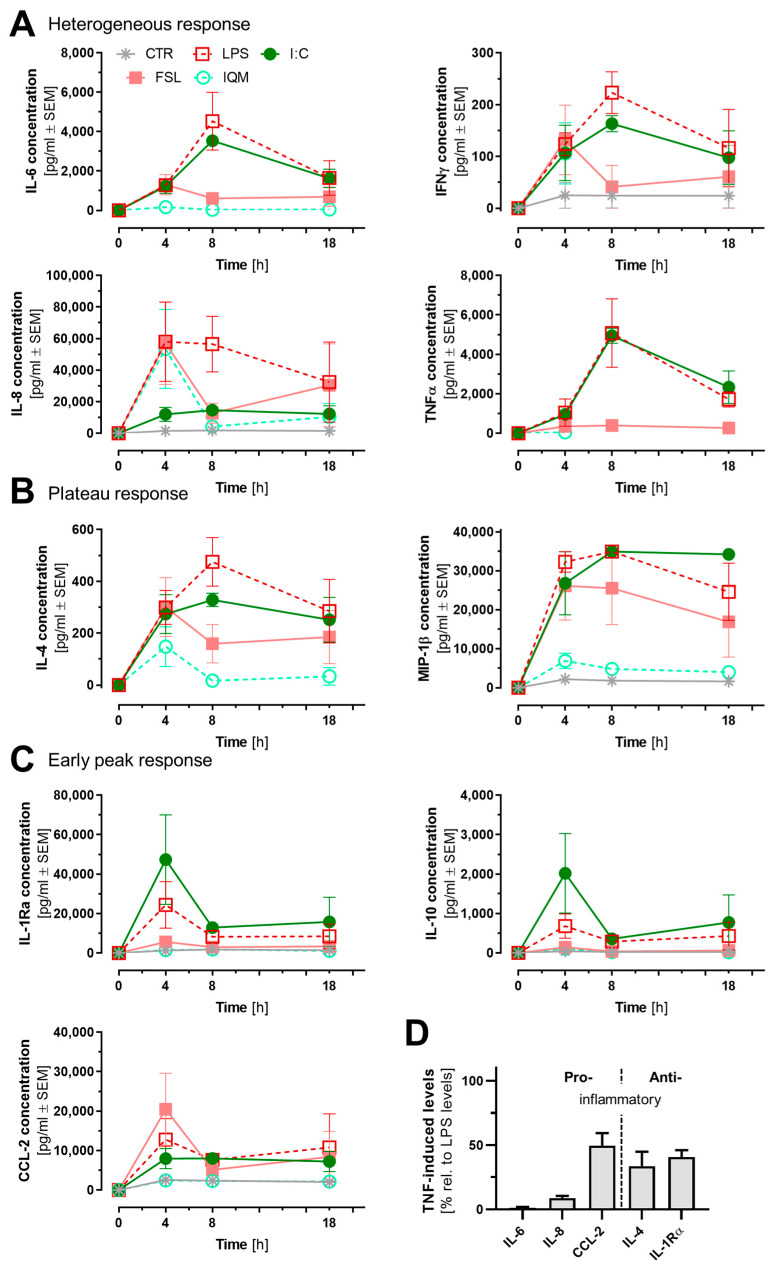
Release of inflammatory mediators from microglia-like cells. DoC7 MGLCs were treated for 4, 8, and 18 h with different stimuli: LPS (100 ng/mL), I:C (10 µg/mL), FSL (100 ng/mL), or IQM (10 µg/mL). Cytokines were measured in cell supernatant by a multiplexed immunoaffinity method. No data are shown when the concentration was below the lower limit of quantification (LLOQ) of 13 pg/mL (IL-6), 66 pg/mL (IFNγ), 38 pg/mL (IL-8), 39 pg/mL (TNFα), 49 pg/mL (IL-4), 16 pg/mL (MIP-1β), 274 pg/mL (IL-1Ra), 44 pg/mL (IL-10), and 149 pg/mL (MCP-1) for all data points under a certain condition (time point and stimulus). If the concentration was not below the LLOQ for all data under a certain condition (time point and stimulus), data below the LLOQ were assumed to be zero. The concentration of IL-8 with LPS at ≥4 h was above the upper limit of quantification (ULOQ) of 83,160 pg/mL. Also, the concentration of MIP-1β was above the ULOQ of 34,930 pg/mL. Therefore, these data points are shown. Data are means of the biological replicates ± SEM (N = 3). Inflammatory mediators are grouped based on cytokine release profiles as follows: (**A**) heterogeneous responses, (**B**) plateau responses, and (**C**) early peak responses. (**D**) DoC7 MGLCs were treated for 8 h with TNF (10 ng/mL). Cytokine levels were measured and normalized to the levels triggered by LPS (100 ng/mL). For instance, TNF triggered much less (>30 fold) IL-6 than LPS, while CCL-2 production differed only by a factor of two. Three pro-inflammatory and two anti-inflammatory cytokines are shown (N = 3).

**Figure 5 cells-14-01687-f005:**
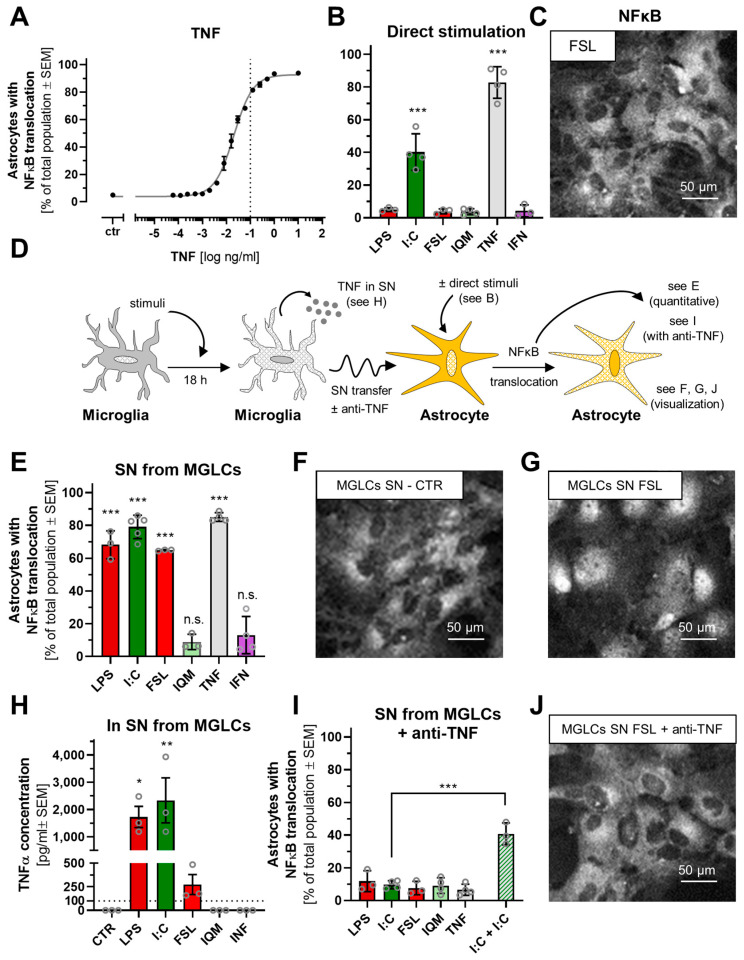
Differential astrocyte activation by microglia-like cells exposed to different inflammogens. (**A**) Three days after seeding, astrocytes were treated with different concentrations of TNF (highest: 10 ng/mL). After 60 min of treatment, astrocytes were fixed and immunostained against the NFκB subunit p65. Data are shown as the percentage of cells with a nuclear NFκB p65 localization [17]. Data are means of the biological replicates ± SEM (N = 4; n = 3). (**B**) Quantification of NFκB translocation in astrocytes after treatment with LPS (100 ng/mL), I:C (10 µg/mL), FSL (100 ng/mL), IQM (10 µg/mL), or TNF (10 ng/mL) for 60 min. Data are means of the biological replicates ± SEM (N = 3–4; n = 3). (**C**) Representative immunofluorescent image showing the NFκB p65 signal in astrocytes treated with FSL-1. The nuclear counterstain (Hoechst-33342) is not shown here but is illustrated in Appendix A (it localizes where the p65 stain shows holes). (**D**) Schematic representation of the experimental protocol used for the supernatant (SN) transfer from stimulated MGLCs to astrocytes. (**E**) The SN from MGLCs treated for 18 h with the indicated stimuli was transferred to astrocytes. After 60 min, NFκB translocation was quantified in astrocytes. Some further controls are shown in Appendix A. Data are means of the biological replicates ± SEM (N = 3–4; n = 3). (**F**,**G**,**J**) Representative immunofluorescent images showing the NFκB p65 signal in astrocytes treated with SN from MGLCs treated with medium, FSL-1, or FSL-1 plus the anti-TNF antibody. The nuclear counterstain (Hoechst-33342) is not shown here but is illustrated in Appendix A. (**H**) The TNF concentration was determined by immunoaffinity analysis in the MGLC medium after 18 h of treatment with the indicated stimuli. Data are means of the biological replicates ± SEM (N = 3). (**I**) The SN from MGLCs treated for 18 h with the indicated stimuli was harvested and then incubated with an anti-TNF antibody. This cocktail was transferred onto astrocytes for 60 min. NFκB translocation in astrocytes was quantified. In the “I:C + I:C” condition, fresh I:C (10 µg/mL) was re-added to the I:C + SN during incubation of SN with anti-TNF antibody. Data are means of the biological replicates ± SEM (N = 3–4; n = 3). For the potency assessment of the anti-TNF antibody, see Appendix A. Significance was evaluated by an unpaired *t*-test *** *p* < 0.001. Significance was evaluated by ANOVA followed by Dunnet’s post hoc test (relative to medium of unstimulated MGLCs) if not stated otherwise. * *p* < 0.05, ** *p* < 0.01, *** *p* < 0.001, n.s. = not significant.

## Data Availability

Raw data have been deposited on Zenodo and are publicly accessible under the following DOI: 10.5281/zenodo.17199104 [72].

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
