# Peer review of "Differential Responses of Human iPSC-Derived Microglia to Stimulation with Diverse Inflammogens"

_cells, 2025, doi:10.3390/cells14211687_

Round 1
Reviewer 1 Report
Comments and Suggestions for Authors
The authors systematically and meticulously analyzed the responses of human induced pluripotent stem cell (iPSC)-derived microglia-like cells (MGLCs) to various inflammatory stimuli.
The authors logically substantiate their hypotheses and, through a comprehensive presentation of relevant results, articulate their arguments.
A particularly noteworthy aspect of the authors' study is their exhaustive examination of the multifaceted cell morphologies and cytokines expressed by differentiated MGLCs in response to diverse stimuli. This aspect is likely to captivate the interest of readers.
However, further consideration is necessary when analyzing the inflammatory response of differentiated astrocytes to various stimuli. While the involvement of astrocytes in neuroinflammation is evident, concurrent analysis of alterations in neurotrophic factor release and inflammatory cytokine secretion is imperative to accurately assess the inflammatory response to each stimulus, taking into account the cell's intrinsic function. Additionally, it is evident that the omission of any mention of gliosis in the context of neuroinflammation, driven by astrocytes, constitutes a significant lacuna in the discussion. Consequently, the incorporation of observations pertaining to vimentin or GFAP would serve to enhance the study's appeal and credibility for the intended audience.
Author Response
For “Differential responses of human iPSC-derived microglia to the stimulation with diverse inflammogens”
Responses to Reviwer
Reviewer 1
Comment 1:
The authors systematically and meticulously analyzed the responses of human induced pluripotent stem cell (iPSC)-derived microglia-like cells (MGLCs) to various inflammatory stimuli.
The authors logically substantiate their hypotheses and, through a comprehensive presentation of relevant results, articulate their arguments.
A particularly noteworthy aspect of the authors' study is their exhaustive examination of the multifaceted cell morphologies and cytokines expressed by differentiated MGLCs in response to diverse stimuli. This aspect is likely to captivate the interest of readers.
Response 1: We thank the reviewer for the positive feedback and appreciation of our work.
Comment 2: However, further consideration is necessary when analyzing the inflammatory response of differentiated astrocytes to various stimuli. While the involvement of astrocytes in neuroinflammation is evident, concurrent analysis of alterations in neurotrophic factor release and inflammatory cytokine secretion is imperative to accurately assess the inflammatory response to each stimulus, taking into account the cell's intrinsic function.
Response 2: We agree that a more extensive study of astrocyte responses would be valuable. The focus of the present study was on microglia-like cells (MGLCs): Astrocytes were used here as “biosensor” to demonstrate the functional relevance of mediator cocktails released from MGLCs. Nonetheless, we appreciate this insightful suggestion and have incorporated a statement in the conclusions highlighting the importance of analyzing astrocyte-specific changes in different contexts (e.g. different levels of growth factors or cell stressors) (p. 24, l. 933fff).
Comment 3: Additionally, it is evident that the omission of any mention of gliosis in the context of neuroinflammation, driven by astrocytes, constitutes a significant lacuna in the discussion. Consequently, the incorporation of observations pertaining to vimentin or GFAP would serve to enhance the study's appeal and credibility for the intended audience.
Response 3: We have now addressed this important point, similar to comment 2, in the conclusions (p. 24, l. 933fff). In line with the interesting suggestions by the reviewer, we are currently working on a more complex MGLC-astrocyte interaction model in 3D. This may also reveal an equivalent of scarring responses.
Reviewer 2 Report
Comments and Suggestions for Authors
Wolfbeisz et al. characterize the diverse activation states of human iPSC derived microglia-like cells (MGLCs) using a combination of transcriptional signatures, morphology and cytokine secretion profile. Microglia are the primary immune cell types in the brain, but information on the dynamics of their response to different immunogenic challenge is missing or mostly available in non-human contexts. The paper is really well written, the methods have been described elaborately, and supplementary information is extensive. The authors have done a really good job overall, so I do not have any major experiments to suggest for the revision. However, the manuscript will be more contextual if they can summarize their findings in the context of what is known through mouse and human monocyte-to-macrophage studies in the conclusion. Two other minor points would be to include that all these data are generated from a single iPSC line, so it does not capture the biological variability in these processes (this should be discussed as a limitation of the study) and clarify which of the data points are from independent experiments versus technical replicates. The authors have attempted this but currently, it is not consistent across all the figures. One way to do this would be to put N= number of independent EBs/ factories, n= number of harvests or cultures set up from the same factory. Some other minor concerns are listed below.
- There are a few typographic errors such as in line 57 and 591.
- Gene names must be italicized.
- 1I: 12h should be removed from the X-axis as data was not collected at this time point. Also, the legend should say gray stars are control below the LPS and I:C, like in Fig.4.
- 1D: were the data points generated using cells harvested from the same PreMacs culture or each point is from an independent culture/ factory?
- 2A: IFNGR, not INFGR. Also, better to replace cell inside with cytoplasm.
- 2B is not marked in the legend on line 661.
- Was the direction of change in gene expression considered for Fig.2E? It is not clear from the legend.
- Line 724: Including the cell viability data in the supplementary will strengthen this observation. What was the assay used for measuring viability?
- Line 739: “This may be due to only a subpopulation expressing the respective receptors.” The authors can test this hypothesis in their transcriptome data.
- 3E: it is always a better idea to represent phosphorylation changes as a percentage of the total protein. Did the authors check if STAT1/ STAT3 levels change due to IFN treatment in MGLCs?
- 5E: why is the control bar missing? What is the effect of SN from control/ untreated MGLCs on NF-kB nuclear translocation in the astrocytes? It is shown in 5F, but the quantification is missing in 5E.
- Line 215: how were the concentration of various inflammogens selected for treatment? If these are based on prior studies, those should be cited here.
- S3: how do you explain the pattern of GFAP expression? Is this observation concordant with previous mouse studies?
- S4 legend: where are the red boxes?
Author Response
For “Differential responses of human iPSC-derived microglia to the stimulation with diverse inflammogens”
Responses to Reviewer
Reviewer 2
Comment 4: Wolfbeisz et al. characterize the diverse activation states of human iPSC derived microglia-like cells (MGLCs) using a combination of transcriptional signatures, morphology and cytokine secretion profile. Microglia are the primary immune cell types in the brain, but information on the dynamics of their response to different immunogenic challenge is missing or mostly available in non-human contexts. The paper is really well written, the methods have been described elaborately, and supplementary information is extensive. The authors have done a really good job overall, so I do not have any major experiments to suggest for the revision.
Response 4: We are grateful for the reviewer’s positive feedback. We are glad to hear that our comprehensive characterization and documentation is appreciated.
Comment 5: However, the manuscript will be more contextual if they can summarize their findings in the context of what is known through mouse and human monocyte-to-macrophage studies in the conclusion.
Response 5:
As we wanted to keep the conclusion brief, we have addressed these points throughout the “results&discussion” section. There is a lot of information in authoritative databases and the review literature on differentiation markers for microglia and respective precursor states (e.g. PreMacs, MGLCs). We extracted this information from the literature/databases and put our data in this context. For this, refer to p.13, l. 576-581 with the following text passage: “To further evaluate lineage identity, we selected markers typically expressed in microglia, macrophages, lymphoid cells, astrocytes, and neurons. Notably, microglia-associated genes such as AIF1 (encoding IBA-1), CX3CR1 (the IL-34 receptor critical for microglial survival), and P2RY12 (a microglia-specific marker) were upregulated in DoC7 MGLCs (Figure 1G, S3). Additional microglial identity markers [51,56], including MERTK, and OLFM3 were upregulated over time.” Another text passage is found on p.13, l.581fff: “The results of single sample gene set enrichment analyses (ssGSEA) showed that genes related to a macrophage/microglia identity were selectively upregulated. For comparison, gene sets related to hepatocytes (other organ) or astrocytes (other brain cell type) were examined in similar ways and found not to be enriched (Figure 1F).”
We characterized not only cells in their (resting) ground state, but also checked whether the inflammatory responses of MGLCs were well-aligned with the literature. In this context, we checked specifically for inflammation responses in microglia, e.g. on p. 14, l. 636ff, we wrote: "Also, I:C evoked a strong overall transcriptional response, consistent with previous reports on a potent I:C-mediated activation of microglia [74,79,80].”. Moreover, on p. 19, l. 800ff, we stated: “We assessed the time-dependent release of mediators from MGLCs after exposure to a panel of inflammatory stimuli. A basal release (from resting cells) was found for MIP-1β, IL-8, IFN, IL-1Ra, IL-10, and CCL-2 (Figure 4A-C). This essentially aligns with previous findings reported by others [51-53].
Following the main theme and focus of our manuscript, we also compared differential responses to stimuli. For instance, we wrote on p. 21, l. 829f: “Stimulation with TNF produced noticeably lower cytokine responses compared to LPS (Figure 4D). This observation is consistent with findings by Knappe, et al. [92].”. In an additional step, we compared the differential, stimulus-dependent responses of MGLC to inflammatory responses in other cells types. We wrote on p. 14, l. 601-610: “This ability to respond stimulus-specifically has been demonstrated before in a study on murine microglia responses to LPS and I:C. Different cytokine release profiles were detected [74]. Similarly, human monocyte-derived macrophages exhibited distinct cytokine secretion in response to different TLR stimuli (including LPS and I:C) [75], and human iPSC-derived astrocytes displayed markedly different transcriptomic patterns following TNF or IFN stimulation [17]. The diversity of activation states of inflammatory cell types has been discussed earlier [14-16], and our initial data indicated that this deserves further exploration for human MGLCs. The distinct, stimulus-dependent responses were therefore further characterized in subsequent experiments.”
Comment 6: Two other minor points would be to include that all these data are generated from a single iPSC line, so it does not capture the biological variability in these processes (this should be discussed as a limitation of the study)
Response 6: We thank the reviewer for highlighting this important limitation. Indeed, all data presented in this study were generated using a single iPSC line, which limits the ability to capture biological variability inherent to different genetic backgrounds and donor-derived cells. This constraint may affect the generalizability of our findings and should be considered when interpreting the results. We have now explicitly acknowledged this limitation in the conclusion section (p. 24, l. 948ff).
Comment 7: ..and clarify which of the data points are from independent experiments versus technical replicates. The authors have attempted this but currently, it is not consistent across all the figures. One way to do this would be to put N= number of independent EBs/ factories, n= number of harvests or cultures set up from the same factory.
Response 7: We thank the reviewer for this important suggestion. We have revised the figure legends and methods section (p. 11, l. 511ff) throughout the manuscript to clearly distinguish between independent biological replicates (N) and technical replicates.
Comment 8: There are a few typographic errors such as in line 57 and 591.
Response 8: We have verified line 57 and corrected the identified typographic errors. However, we could not locate line 591. (please kindly confirm the reference or provide additional context so that we can address this point accurately.)
Comment 9: Gene names must be italicized.
Response 9: Thank you for highlighting this important formatting detail. We have carefully reviewed the manuscript and italicized all abbreviated gene names (gene symbols).
Comment 10: In figure 1I: 12h should be removed from the X-axis as data was not collected at this time point. Also, the legend should say gray stars are control below the LPS and I:C, like in Fig.4.
Response 10: Thank you. These points are absolutely well-taken. We corrected these accordingly.
Comment 11: In figure 1D: were the data points generated using cells harvested from the same PreMacs culture or each point is from an independent culture/ factory?
Response 11: As addressed in Comment 7, we clarified now the issue with the inconsistency regarding the naming of replicates. In the case of 1D, PreMacs from four independent factories were harvested and MGLCs were differentiated.
Comment 12: 2A: IFNGR, not INFGR. Also, better to replace cell inside with cytoplasm.
Response 12: Thank you. We corrected these points accordingly.
Comment 13: 2B is not marked in the legend on line 661.
Response 13: Thank you. This was updated in the manuscript.
Comment 14: Was the direction of change in gene expression considered for Fig.2E? It is not clear from the legend.
Response 14: The direction of gene expression change was not considered for Figure 2E. Differentially expressed genes were defined by an absolute logâ‚‚ fold change ≥ 1 and an adjusted p-value < 0.01 (Wald test with Benjamini-Hochberg correction), including both up- and downregulated genes. Overlap between differentially expressed gene sets was calculated using the GSVA package (v2.07) as the proportion of overlapping genes relative to the smaller gene set.